# Full-Scale Field Test on Construction Mechanical Behaviors of Retaining Structure Enhanced with Soil Nails and Prestressed Anchors

**Hui Wang [1], Jianhua Cheng [1,\*], Hujun Li [1] , Zhilin Dun [1] and Baoquan Cheng [2]**

1   School of Civil Engineering, Henan Polytechnic University, Jiaozuo 454000, China;
    wanghui9962@hpu.edu.cn (H.W.); lihujunsr@163.com (H.L.); dzl1964@163.com (Z.D.)
2   School of Civil Engineering, Central South University, Changsha 410075, China; curtis_ch@163.com
*   Correspondence: cheng15@hpu.edu.cn

**Abstract:** Soil nailing combined with prestressed anchors has a good workability and is relatively cheap in constraining the horizontal displacement. Current research on the technique, whether theoretical analyses, numerical simulations, or model tests, was conducted under ideal working conditions. However, in fact, external disturbances, such as tensioning-lagging of the anchor, are very common and play an important role on stress and displacement. Therefore, it is of great significance to carry out a field test considering the effects of external disturbances, which can obtain real and reliable data through real-time monitoring. In this paper, the impacts of the construction conditions on practical engineering are discussed based on in situ tests, and some reasonable suggestions for the upgrading of misbehaviors in the current construction situation are put forward. In particular, the influence features of soil predisturbance, excessive excavation, unloading on the surface of edges, tensioning-lagging of the anchor, and continuous rainfall on the stress–time curve of soil nails under practical working conditions are analyzed. Behaviors of three different retaining structures enhanced with (i) soil nails; (ii) soil nails and prestressed anchors without unbonded parts; and (iii) soil nails and prestressed anchors with a 2.5 m unbonded part were monitored during staged excavation to investigate the influences of (i) the prestressing force and (ii) the unbonded part of the prestressed anchors on the performance of the entire retaining system. Results show that (i) the prestressing force is the main factor affecting the stress and deformation of the composite retaining system, which is consistent with the existing literature; (ii) the variation of the magnitude and distribution of the soil nail force responding to the anchor prestressing force, however, showed no systematic trend; and (iii) the unbonded part of anchors, which was validated to be the main factor affecting the structural stability in dense materials in the existing literature, is found to have a minor influence in loose fill materials used in this study.

**Keywords:** retaining structure; soil nailing; prestressed anchor; in situ test; mechanical behavior

## 1. Introduction

Soil nailed retaining structures are widely used in slope supporting, tunnel retaining, and as bracing structure for foundation pits to limit deformation and improve safety. The advantages of soil nailing include its simple structure, flexible operation, and high cost-effectiveness [1–3]. Soil nails have a high ratio of circumference to cross-section area, and therefore rely essentially on friction resistance for the load transfer. The soil nail force is transferred to the surrounding soils through soil–grout interface friction, which arises from the relative displacement between the soil and the grout. The shear strain required to mobilize the nail force is very small; therefore, soil nails usually start to function soon after the excavation when deformation occurs.

Soil nailing is one of the in situ soil reinforcement techniques which are widely utilized by civil engineers in underground construction, slope stability, housing construction on

inclined or vertical cuts, etc. A vast amount of research has been conducted on the retaining structure. As for in situ or laboratory tests, some studies were focused on investigating the pullout behavior of pressure-grouted soil nails [4–7]. Numerical models were established to examine the contribution of soil nails towards the slope stability [8–11]. However, the soil conditions in which soil nailing can be applied are limited and the deformation is difficult to control. Therefore, for projects in urban areas, a soil nail–prestressed anchor composite retaining structure is preferable. This is because the additional prestressed anchors can assist, in addition to the soil nails, to limit the displacement of the slope surface [12–15]. A typical layout of the soil nail–prestressed anchor composite retaining structure is shown in Figure 1.

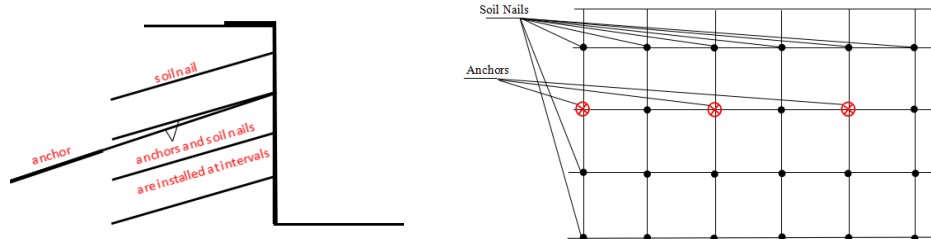

**Figure 1.** Vertical section of soil nail–prestressed anchor composite retaining structure.

In such a composite retaining structure, the combined contribution of soil nails and prestressed anchors is expected to provide higher resistance against rainfall or overload-induced failure [16]. In the existing literature, the interaction mechanism between soil nails and prestressed anchors is investigated mostly based on numerical studies [17]. Very limited model and in situ test results [18–20] and analytical theories and methods were reported despite the widespread use of such retaining structures. This is due to (i) model and in situ tests being expensive, complicated, and time-consuming to conduct. For example, many delicate instruments are needed, such as stress and strain gauges which are prone to damage during testing and installation; (ii) in situ tests may disturb the scheduled construction process and the in situ soil layers. In addition, previous studies were mainly focused on the behavior of such retaining structures installed in dense materials [21]; the corresponding behaviors in loose fill materials need to be further investigated.

In this paper, a full-scale in situ test on a composite soil nailed retaining structure enhanced with prestressed anchors installed in loose materials was carried out to investigate the mechanical behavior of soil nails and prestressed anchors during excavation. Great precaution was taken during the in situ tests to avoid possible damages to the instruments. For the installed instruments, obvious signboards were set next to them, and the staff on site were reminded to avoid rolling over, covering of, and collision with the monitoring instruments. In addition, daily inspection was strengthened and replacements were done immediately in case of damage or abnormality. Therefore, the survival ratio of gauges was ensured to be as high as 99%. A total of 90 strain gauges were installed, and only one was damaged. The effects of the prestressing force and unbonded length of the prestressed anchors on the nail forces were investigated. Scientific problems to be solved include: (i) the effects of prestressing force on the nail forces; (ii) the effects of the unbonded part of prestressed anchors on the nail forces; and (iii) the differences between soil nails and prestressed anchors in terms of construction mechanical behaviors in loose fill materials. The research presented in this paper is useful for the rational design and serviceability analysis of composite retaining structures.

## 2. Site Conditions

The tested foundation pit was designed for the Zhengzhou University Science and Technology Building which consists of a 20-story tower building, a 3-story skirt building, and 1-story basement. The depth of the foundation pit was 6.53 m. A soil nail–prestressed anchor composite retaining structure was used to support the north wall of the foundation

pit where underground pipelines concentrate, and therefore, strict deformation control was required. A soil nailed retaining structure was used to support the south wall of the foundation pit. Field investigation showed the ground water table was stable and beneath the foundation pit at a depth ranging between 10.1 and 10.9 m. Due to this reason, the influence of ground water on the retaining structure could be safely ignored. The layout of the foundation pit is shown in Figure 2.

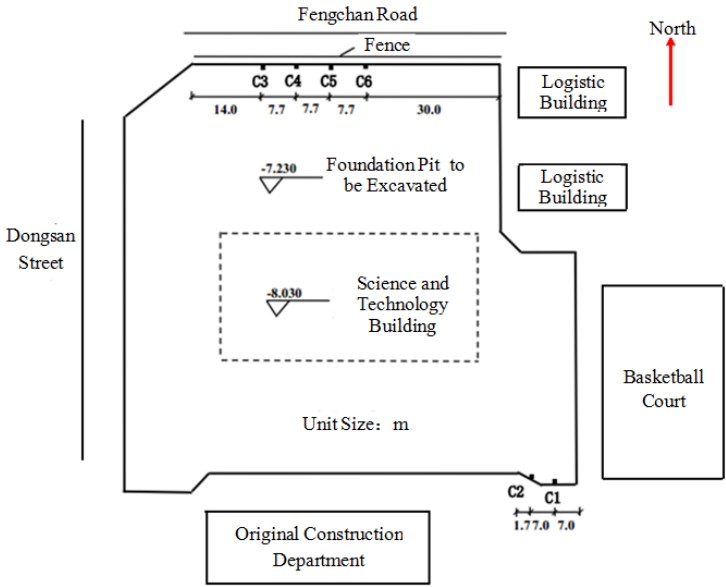

**Figure 2.** The layout of the foundation pit.

There are a series of testing sites noted as C1 through C6. C1 and C2 are soil nail retaining structures located in the south; C3 through C6 are soil nail–prestressed anchor composite retaining structures located in the north, among which there are no unbonded parts in the anchors in C3 and C4 whereas there are 2.5 m unbonded parts in the anchors in C5 and C6. For C2, a cement mixing zone was located near the testing site; therefore, unloading of cement and vibration of the cement mixer may affect the development of soil nail force during staged excavation. In addition, there were two deserted holes near the north wall, left for confirming the actual position of the gas pipelines before excavation. For C3 and C6, which were located near the two holes, previous disturbance may have changed the magnitude and distribution of the soil nail force. For revealing the mechanical behavior of the composite soil nailed retaining structure, the test results derived from testing sites C2, C3, and C6 were discarded, and the other data from C1, C4, and C5 were taken for analysis.

### 3. Soil Properties

The in situ soil layers from top to bottom were (i) silt, 2.20 m; (ii) silty clay, 2.30 m; (iii) silt, 1.10 m; and (iv) silty clay, 2.30 m. The properties of every soil layer are listed in Table 1.

**Table 1.** The mechanical parameters of the soil layers.

| Soil Nail | Depth (m) | Length (m) | Inclination (°) | Spacing (m) |
|---|---|---|---|---|
| 1 | 1.20 | 9.00 | 10 | 1.50 |
| 2 | 2.70 | 9.00 | 10 | 1.50 |
| 3 | 4.20 | 9.00 | 10 | 1.50 |
| 4 | 5.70 | 7.00 | 10 | 1.50 |

## 4. Construction of the Retaining Structure

For the soil nail–prestressed anchor composite retaining structure, soil nails and prestressed anchors were distributed in a "square" layout with an equal vertical and horizontal spacing of 1.4 m. Boreholes with a diameter of 120 mm and an inclination of 10° were predrilled manually. After the installation of the steel reinforcement bars into the boreholes, two-stage grouting was applied. In stage one, conventional gravity grouting was used to seal the annular space between the steel bar and the hole with cement grout. In stage two, a predefined length of the soil nail was grouted using a grout pressure of approximately 1.5 MPa. The objective of the pressured grouting was to fabricate the annular cement grout and permeate the surrounding soil. Two-stage grouting has been used successfully to reinforce cut slopes, excavations, tunnels, etc. to increase the performance of soil nails and therefore reduce the number of required soil nails in many countries and areas [22–24].

Each soil nail used in the experiment consists of a ribbed steel reinforcement bar of 18/22 mm diameter, the elastic modulus of which is 200 GPa. The soil nails were embedded in a grout mixture with a water/cement ratio of 0.5. The facing was made up of 200 mm by 200 mm thin steel mesh grids (6 mm in diameter). The facing was enhanced with two reinforcement bars (12 mm in diameter) in both horizontal and vertical directions. The detailed enhancement configuration is shown in Figure 3.

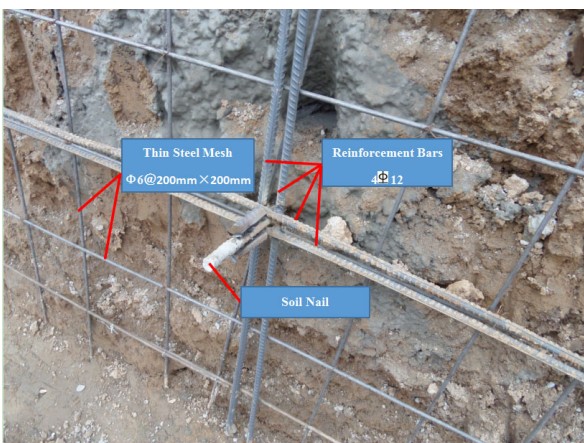

**Figure 3.** The enhancement configuration of the facing.

The concrete slurry, with a thickness of 80 mm and a compression strength of 20 MPa, was sprayed to fill the gap between the steel mesh and the soil behind and form the retaining wall. A steel plate, which was 200 mm in length, 200 mm in width, and 20 mm in thickness, was installed at the conjunction between the prestressed anchor and the retaining wall to reduce the possible stress concentration. The designed value of prestressing force of the anchor was 50 kN. The design parameters of the soil nails and prestressed anchors in different retaining structures are summarized in Tables 2 and 3.

**Table 2.** Design parameters of soil nailed retaining structure.

| Soil Layer | D (m) | $\gamma$ (kN/m$^3$) | $c$ (kPa) | $\varphi$ (°) | $\tau$ (kPa) |
|:---:|:---:|:---:|:---:|:---:|:---:|
| ① | 2.20 | 18.1 | 14.0 | 20.0 | 52.0 |
| ② | 2.30 | 17.9 | 20.0 | 15.0 | 50.0 |
| ③ | 1.10 | 18.2 | 15.0 | 21.0 | 60.0 |
| ④ | 2.30 | 18.2 | 21.0 | 16.0 | 56.0 |

**Note**: D is thickness, $\gamma$ is unit weight, $c$ is cohesion, $\tau$ is shear stress of soil/grout interface, and $\varphi$ is internal friction angle.

**Table 3.** Design parameters of composite soil nailed retaining structure.

| Soil Nail/ Anchor | Depth (m) | Length (m) | Bonded Length (m) | Spacing (m) |
|---|---|---|---|---|
| 1 | 1.20 | 9.00 | - | 1.50 |
| 2 | 2.70 | 12.00 | 12.0/9.5 | 1.50 |
| 3 | 4.20 | 9.00 | - | 1.50 |
| 4 | 5.70 | 7.00 | - | 1.50 |

## 5. Fabrication of Testing Instruments

In the in situ tests, the soil nails were instrumented with vibrating wire strain gauges (JMZX-416A), which were attached to the steel tendon of each soil nail. JMZX-416A was applied to measure the stress of stressed reinforcement in reinforced concrete structures, the measuring range and sensitivity of which are 200 MPa and 0.1 MPa, respectively. Readings were obtained with and stored on a data logger (JMZX-3001). During the installation of the strain gauges, great caution was undertaken to protect the gauges and ensure their survivability. The fixed strain gauge consists of four parts: sensor, connecting rod, wire, and plug. Details of the strain gauge are illustrated in Figure 4.

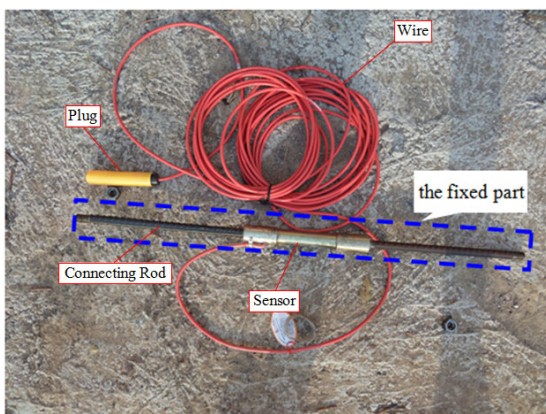

**Figure 4.** The details of the strain gauge.

Before gauge installation, their initial readings were brought to the prescribed range. Gauges were then mounted onto the blocks used for fixation and wrapped with a plastic cover to isolate and protect them from subsequent grouting. A typical installation procedure of the strain gauge is specified as follows. First, the steel bar is cut into the predesigned size and length. Then, the fixed part of the strain gauge, as shown in Figure 4, is welded to the steel bar. It is noted that the overall length of soil nail remains unchanged after the installation of the strain gauge. Afterward, wires are arranged along the length of the soil nail and fixed on the soil nail with waterproof tape. Finally, wires are collected at the top of the slope and connected to data loggers. It is worth noting that care must be exercised during welding to avoid sensor damage, and the part where the sensor and the connecting rod are combined should be covered with a wet cloth and moisturized constantly to avoid overheating. Afterwards, fiber optic sensors are fixed onto the soil nail according to the circuit of U. The soil nails, with centralizers at a 2.0 m interval, are placed in the boreholes, followed by two-stage grouting (i.e., gravity and pressure grouting). Figure 5 shows details of the instrumented soil nails.

Different from soil nails, each anchor (25 mm ribbed high yield steel bar) was instrumented with a vibrating wire load cell (MJ-101) at the head to monitor the anchor force, with the exception of strain gauges. Figure 6 presents the details of a load cell.

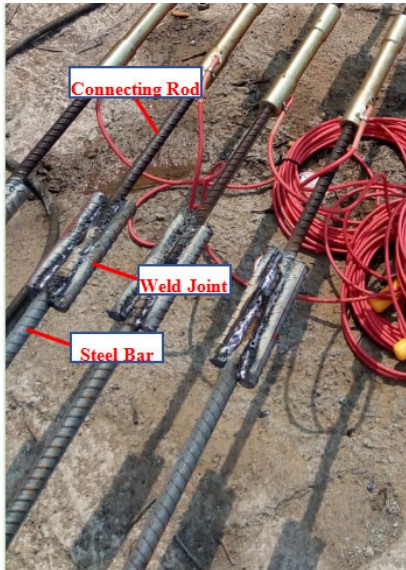

**Figure 5.** The manufactured testing soil nails.

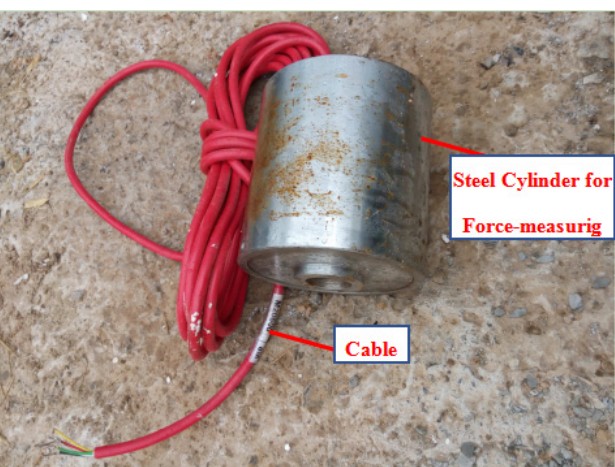

**Figure 6.** The details of the load cell.

Due to the anchor tendons being made of steel bars instead of steel strands or wires, the main difference between soil nail and anchor is the processing of the unbonded part. Depending on the designed unbonded length, after grease was spread, a PVC-corrugated pipe was used to wrap the steel reinforcement bar. The connection between the unbonded and the bonded part was fastened with fine steel wires. Figure 7 presents the manufactured testing anchors.

To investigate the mechanical behavior of composite soil nailing, a series of tests was carried out under different conditions. The measured results from the testing profiles of C1, C4, and C5 were analyzed to study the effects of two important characteristics, namely, prestressing force of the anchor and unbonded length of the anchor, on the soil nail force. For the first parameter, it is worth noting that loss of prestressing force should be avoided if possible. For the second parameter, it can be expressed in terms of the length ratio $Lu/L$, where $Lu$ is the length of unbonded part and $L$ is the entire length of the anchor. Testing results of profiles C1, C4, and C5 were simplified with No.1, No.4, and No.5 in the following analyses. As shown in Table 3, two different unbonded lengths of anchors were used. No.1 represents a simple soil nailed retaining structure; No.4 represents a composite soil nailed retaining structure enhanced with prestressed anchor without unbonded parts; and No.5 represents a composite retaining structure with a 2.5 m unbonded length. The

layout of instruments is shown in Figures 8–10. As can be seen from these diagrams, load cells were installed at the anchor heads to monitor the anchor force and strain gauges which were adhered to the steel bars used to measure the axial strain of the soil nails at different locations. It is important to note that the connection between the soil nails and the facing was robust enough to function properly throughout the entire processes of excavation.

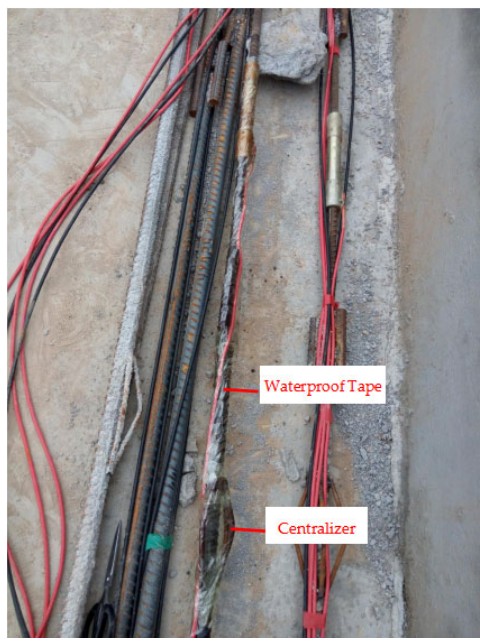

**Figure 7.** The manufactured testing anchors.

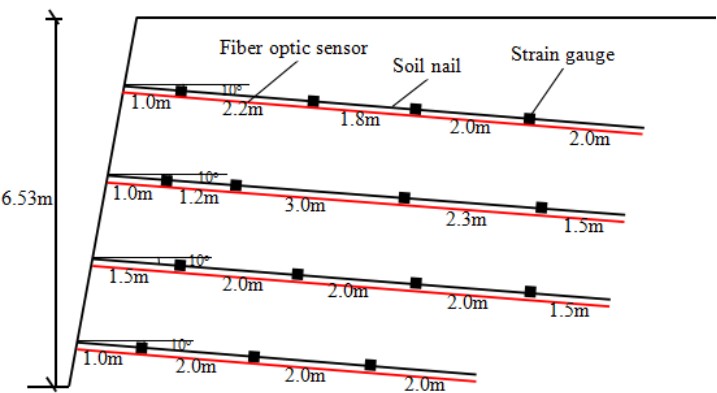

**Figure 8.** Location of instruments for testing set No.1.

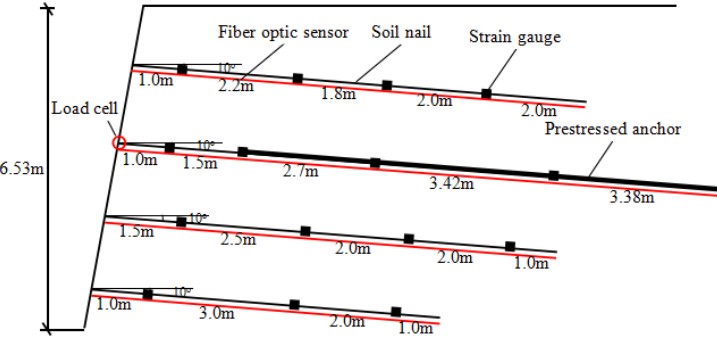

**Figure 9.** Location of instruments for testing set No.4.

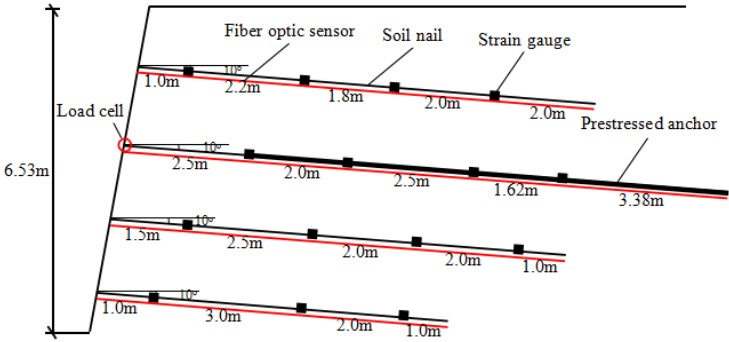

**Figure 10.** Location of instruments for testing set No.5.

## 6. Test Results

### 6.1. Stresses of Soil Nails

After excavation and installation of the soil nails and prestressed anchors, the data acquisition system was established. According to the stress–time curves, the stress on the soil nails increased slowly and tended to be stable after three months. The mechanical behavior of the soil nail–prestressed anchor retaining system was monitored for about three months. Figures 11–15 reveal the stress–date relationship obtained from the in situ tests during the period of three months, in which Tij presents the stress value of the j-th strain gauge calculated from the nail head in the i-th row of soil nails.

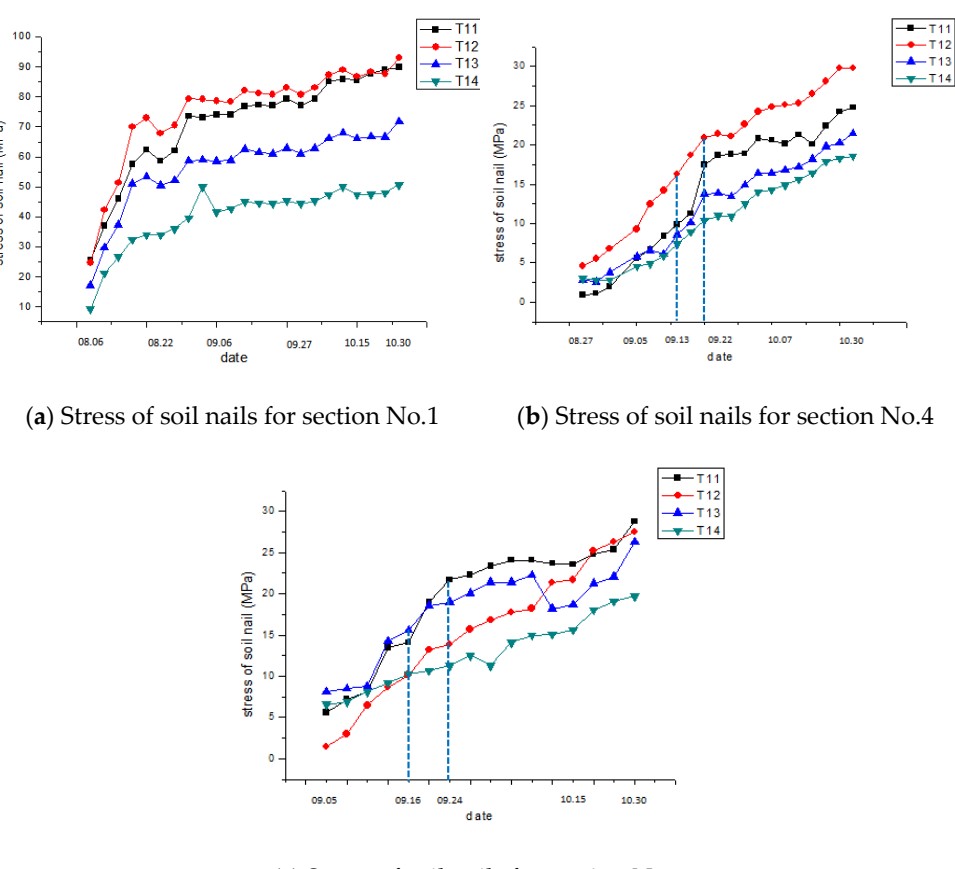

(**a**) Stress of soil nails for section No.1

(**b**) Stress of soil nails for section No.4

(**c**) Stress of soil nails for section No.5

**Figure 11.** Stress–date relationship of the first row of soil nails for testing sections.

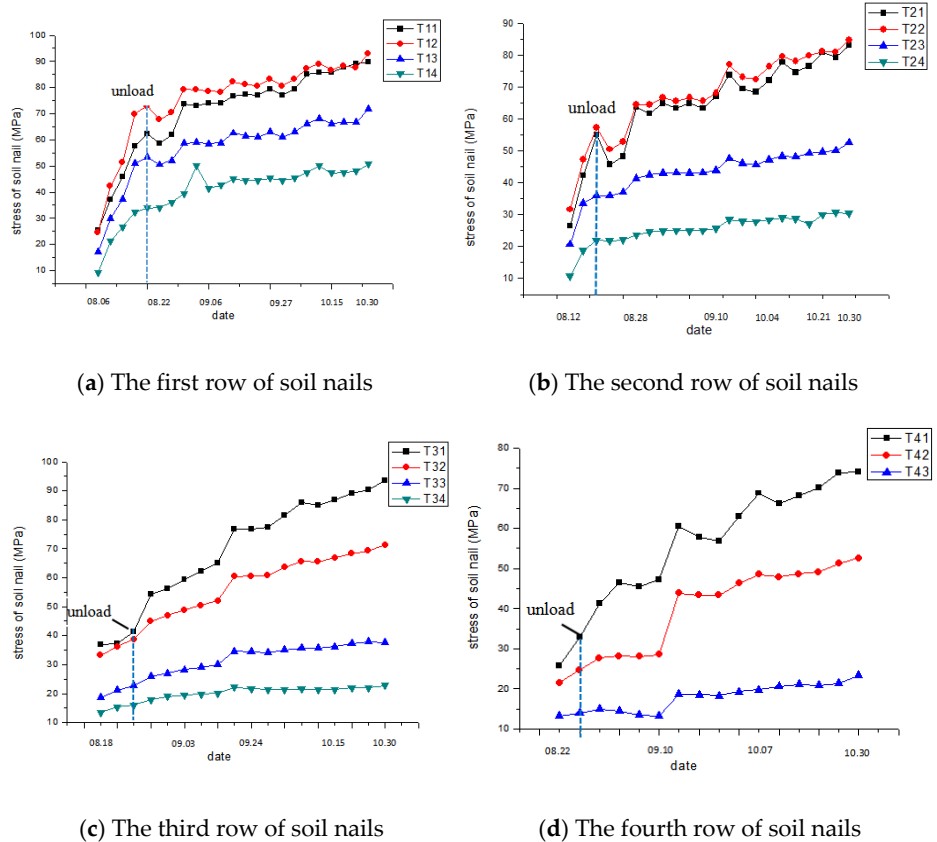

(**a**) The first row of soil nails

(**b**) The second row of soil nails

(**c**) The third row of soil nails

(**d**) The fourth row of soil nails

**Figure 12.** The influence of unloading on stress of soil nails for section No.1.

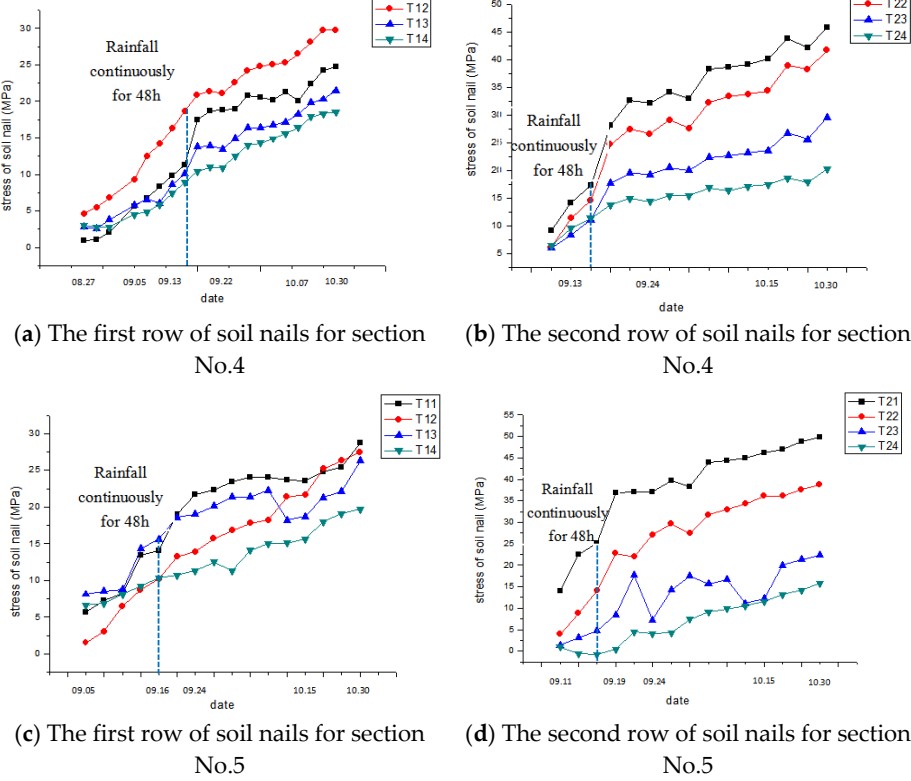

(**a**) The first row of soil nails for section No.4

(**b**) The second row of soil nails for section No.4

(**c**) The first row of soil nails for section No.5

(**d**) The second row of soil nails for section No.5

**Figure 13.** Stresses of the upper two rows for sections No.4 and No.5.

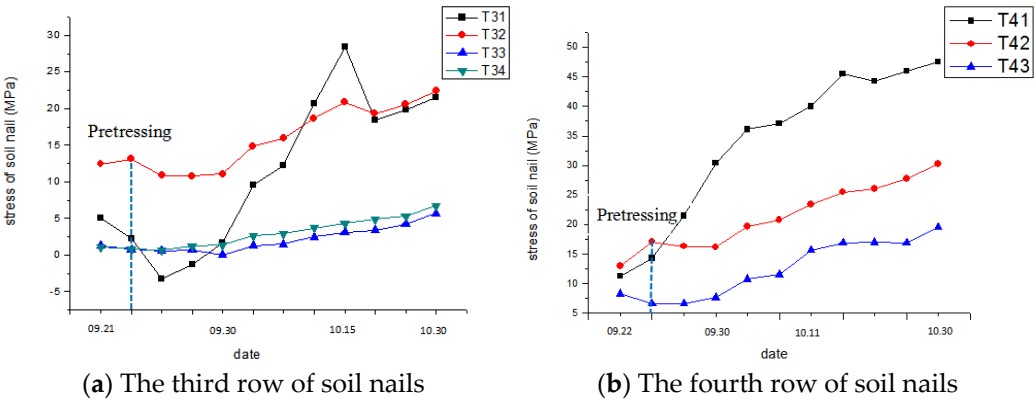

(**a**) The third row of soil nails          (**b**) The fourth row of soil nails

**Figure 14.** Stresses of the lower two rows for section No.5.

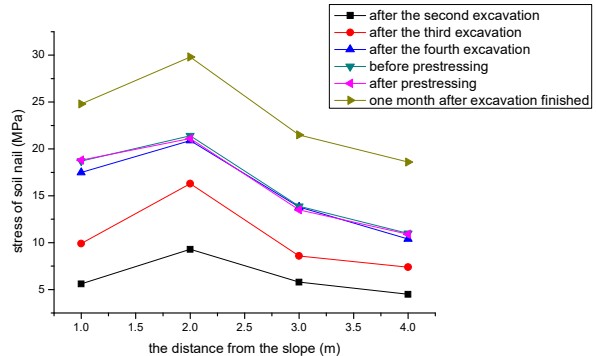

(**a**) Stress of soil nails for section No.4

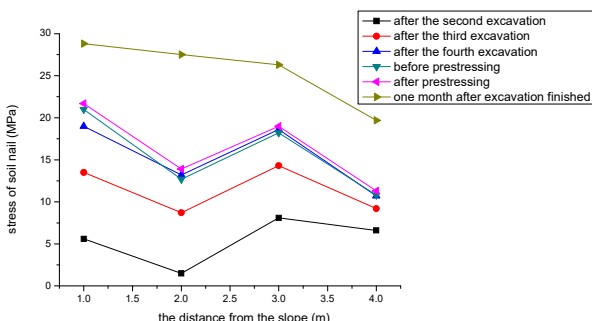

(**b**) Stress of soil nails for section No.5

**Figure 15.** The distribution of nail forces of the first row for sections No.4 and No.5.

Figure 11 presents stresses of the first row of soil nails for sections No.1, No.4, and No.5 during the excavation process. It is evident that there is a non-linear increase in the nail force for section No.1 with each excavation step. The nail force tends towards a stable value when the excavation is completed. This means that there are effects of time and excavation on the internal force of the soil nail. This is consistent with the results reported in the literature [19]. Compared with section No.1, the effect of excavation is less significant for sections No.4 and No.5. The overall stress level monitored from the tests is relatively low. Corresponding to the excavation of the foundation pit, increments of earth pressure due to unloading are transferred to the soil nail through shear stress of soil-grout interface. For this testing project, the foundation pit is adjacent to Fengchan Road in the north, under which a gas pipeline is buried parallel to the foundation wall. The pipeline lies 1.0 m deep and about 1.5 m away from the side of the foundation pit. Due to this reason, the upper soil layers north of the foundation pit for sections No.4 and No.5 had been previously

disturbed, which is considered looser compared to the undisturbed soil for section No.1. Therefore, the friction resistance of the soil/grout interface for sections No.4 and No.5 is lower, and the earth pressure increment transferred to soil nails is lower. According to the above analysis, a conclusion can be drawn that the stress distribution is affected by the density of the soil. The stress level and concentration is higher when the soil is dense; the stress level and distribution is uniform and lower when the soil is loose. The obtained results are consistent with observations reported in the literature [25].

As can be seen from Figure 12, the stress levels of soil nails for section No.1 decrease due to unloading, which only manifests that the first and second rows of soil nails close to the ground surface are predominantly affected; however, the third and fourth rows are less influenced. This is because the soil for section No.1 is dense, and the mechanical property is close to undisturbed soil. The unloading effect from the ground surface cannot reach the lower two rows.

Figure 13 presents the stress history curves of sections No.4 and No.5. The stress distribution and levels are less affected by rainfall, which continuously lasted for 48 h. This is because the infiltration rate of the rainfall in the clay or silty clay is extremely low; for example, in clay soil, the initial infiltration rate is 2.21 mm/min, and the steady infiltration rate is 0.62 mm/min according to the test results [26]. Accordingly, the depth of infiltration is relatively low. In addition, the actual location of the first row of soil nails is moved down to keep away from the gas pipeline, and the exact locations for the two test profiles are $-2.0$ m and $-2.2$ m, respectively.

As can be seen from Figure 14, the stresses of soil nails in the lower two rows decrease heavily compared to the stresses of the upper two rows (as shown in Figure 13). Based on the technical specifications, the bonded tendons can be tensioned effectively when the stress exceeds 15 MPa. However, in fact, the next layer was removed followed by the second excavation as a result of arranged rapid construction; afterwards, tensioning was performed. Consequently, the soil mass influenced by prestressing is in the lower parts instead of the upper parts of the foundation pit. A conclusion can be drawn, that is the stress magnitude and distribution of the soil nails are affected by different prestressing periods: the upper rows are influenced significantly when applying stress earlier; the lower rows will change greatly when applying stress later.

As can be seen from Figure 14, the influence of prestressing on the stress of the third and fourth rows of soil nails is only manifested for some of the nails, which were close to the slope surface, compared to those nails away from the slope surface along the longitudinal axis of the nail, which were affected very little. This shows that the influencing range of prestress is very limited.

As mentioned before, sections No.4 and No.5 are different retaining structures enhanced with (i) soil nails and prestressed anchors without unbonded parts and (ii) soil nails and prestressed anchors with a 2.5 m unbonded length. When comparing the measured results, the influence of the unbonded length on nail forces can be investigated. As can be seen from Figure 15, the main difference lies in the distribution of the nail forces of the first row. For section No.4, without unbonded parts, the distribution of the nail forces is consistent with the documented results, which presents an inverted saddle shape, that is "small in the end and big in the middle". However, for test profile No.5, with a 2.5 m unbonded length, the distribution is manifested as "double peaks", which shows that there may be more than one potential slip surface in loose fill materials. For section No.4, due to the anchor being fixed in the overall length, there is deformation of the steel reinforcement bar to transfer the load from the anchor head to the slope. Accordingly, an anchorage effect cannot function adequately. In other words, it amounts to a longer prestressed soil nail. For the other three rows of soil nails, the difference between the two sections is not so obvious. The effect of the unbonded length, which is considered to be the main reason for slope stability in dense materials, is negligible in loose fill materials.

### 6.2. Distribution of Maximal Nail Forces

The distribution of maximum nail force along the longitudinal axis of the soil nail for sections No.1 and No.4 are shown in Figure 16, in which the broken lines only represent the positions of soil nails, not the lengths.

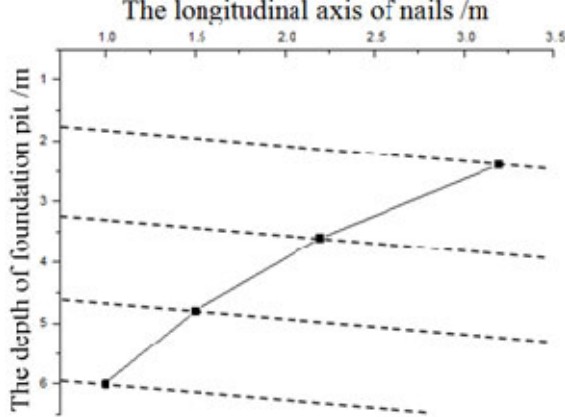

(**a**) The distribution of maximal nail forces along the longitudinal axis of nails for section No.1

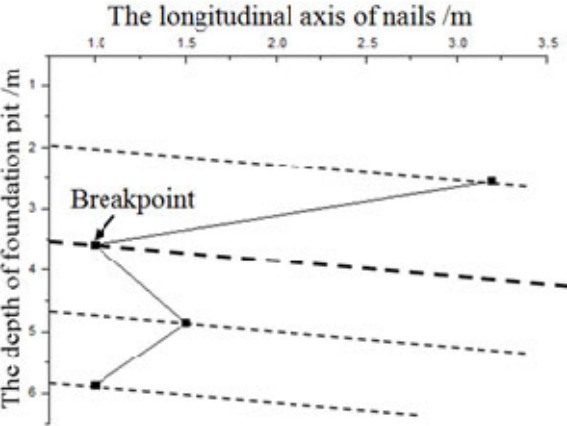

(**b**) The distribution of maximal nail forces along the longitudinal axis of nails for section No.4

**Figure 16.** The distribution of maximal nail forces: (**a**) Section No.1; (**b**) Section No.4.

Figure 16a shows the distribution of maximum nail force along the longitudinal axis of nails for section No.1. From the perspective of a qualitative analysis, this distribution curve of maximal nail force matches with the most critical failure plane of slope. However, the distribution profile of maximal nail forces for section No.4 is different, as can be observed from Figure 16b. There is an obvious breakpoint which indicates that stress distribution is affected by the prestressing. After prestressing, a compressive stress zone arises at the end of the anchor; meanwhile, tensile stress zone appears at the tip of the anchor, where stresses of adjacent soil nails increase according to the superposition principle (Figure 17).

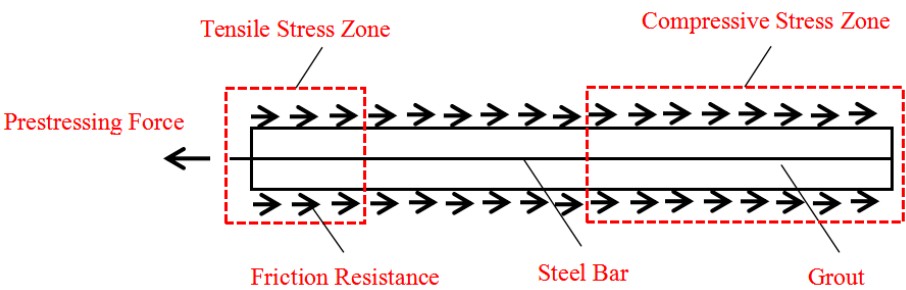

**Figure 17.** Stress distribution of tensioned anchor after prestressing.

In this testing project, the tensile force on the anchor was relatively low. In addition, there is no reaction frame to transfer the load to the slope. Consequently, the anchoring effect cannot function adequately, which makes the effective compressive stress zone limited. Furthermore, the stress zone is isolated instead of connected. The peak stress is not evident for section No.5. The distribution of maximal axial forces of soil nails of the testing profile is not provided. According to Figure 15b mentioned above, the distribution of maximal nail stresses of the first row for section No.5 features an obvious characteristic of "double peaks", which indicates that more than one critical failure plane exists. However, one month after excavation finished, the performance of "double peaks" weakened gradually, due to the creep characteristic of loose fill materials. Therefore, it can be concluded that the development of stress of a single soil nail is neither uniform nor stable, which may be caused by the uniformity of the soil layers.

## 7. Analysis and Discussion

### 7.1. Disturbance Effect

With the progress of foundation pit excavation, the generated load increment causes the relative displacement between the soil and the grout, and the generated interface friction is used to provide the binding force to stabilize the slope. For No.4 and No.5, the soil on the upper part of the foundation pit had been disturbed during the early pipeline construction, which is relatively loose and more compressible than the undisturbed soil. In the loose fill materials, the bonding strength between the soil and the grout decreased, which affected the exertion of friction resistance, and the earth pressure transmitted to the soil nail was relatively smaller, which is consistent with the literature [25].

### 7.2. Sequence Effect

According to the technical specifications, the grout needed to reach 15 MPa before the anchor bolt was tensed. For No.5, due to limitations in the construction period, the construction process was accelerated. The lower layer was excavated soon after the excavation of the second layer, and then the anchor bolt was tensed. This construction method is not conducive to control the deformation of the foundation pit, because the anchor bolt was not stressed when it was not tensed, and the corresponding earth pressure increment distributed to the upper and lower rows of soil nails was large, which increased the horizontal lateral displacement of the foundation pit and weakened the expected effect of the anchor bolt. However, this nonstandard construction method still exists at a large scale in practical engineering. Therefore, the tensioning mode considering the "sequence effect" is of great significance to ensure project safety and achieve economic efficiency.

### 7.3. End Effect

Soil nails are designed according to reinforcement theory. Prestressed anchors are designed as per anchoring mechanism. Composite soil nailed retaining structures enhanced with prestressed anchors should therefore be designed considering the superposition principle. According to the superposition principle, the nail force affected by compressive stress will decrease, while the stresses of soil nails which are adjacent to the tensile stress

zone will increase [27–29]. However, due to the smaller tensile stress, the influence of which is limited, the effect of prestressing only manifests as the decreasing of internal forces of soil nails. For the same soil nail, the reduction range of internal forces is quite different along the length of soil nail, indicating there is an "end effect" for the influence of tensioning on the internal forces of soil nails.

### 7.4. Bimodal Stress

In normal undisturbed soil, with the increase of load, the interface between the grout and the soil will break through the elastic stage, and the interface will slip within a certain range. The load distribution form is significantly changed compared with the complete elastic stage. There is only a sliding friction effect in the sliding section. After the sliding section, it still shows the form of single peak curve due to the relatively smaller axial force. According to the stress distribution curve of each row of soil nails of No.5, there are two stress peaks along the nail length. According to the theory that "the maximum tension of soil nails is near the fracture surface", there may be more than one potential slip surface. This is because the loose fill materials have greater compressibility, looseness, and heterogeneity compared with dense materials, which makes the stress distribution change greatly along the nail length, which is consistent with the research results of the literature [30].

### 7.5. Unbonded Length Effect

Compared with No.5, there is no form of "double peak curve" for No.4, which is the external manifestation of the influence of anchorage on a stress field. For No.5, it is anchored only at the end. The pretension stress applied to the end can be better transmitted to the grout and the soil behind through the elastic deformation of the anchor bolt. The effective range compressive stress is large, and the active support effect is good. For No.4, it is anchored along the full length. The pretension stress applied to the end cannot be effectively transmitted to the soil behind, and the compressive stress formed is mainly concentrated at the end of the anchor.

### 8. Conclusions

In this paper, in situ tests were conducted to investigate the mechanical behavior of soil nail and prestressed anchor enhanced retaining structures. Three types of different retaining structures, including (i) soil nails; (ii) soil nails and prestressed anchors without unbonded parts; and (iii) soil nails and prestressed anchors with a 2.5 m unbonded length, were investigated. The effects of different factors, such as prestressing and unbonded length were discussed. The results obtained may assist practicing engineers in the design of nails and anchors for civil engineering applications. In particular, it was found that:

(i)     The interface friction stress of the soil nail is affected by the density of surrounding soil. The friction stress is higher when soil is denser and harder, which is characterized by stress concentration; the friction stress is lower and uniform when soil is disturbed or looser, which is consistent with theoretical hypotheses;

(ii)    The distribution pattern of soil nail strain along the full length in disturbed soil is bimodal, which indicates that there are two potential slip surfaces, and it is inferred that there are even more. This is one of the important characteristics of disturbed soil different from general clay;

(iii)   The influence of prestressing on the internal force of soil nails is governed by tensioning methods. When the anchor is tensed synchronously, its ability to limit the deformation is obvious, and the upper soil nails are affected. When the anchor is tensed later, the lower soil nails are affected;

(iv)    The prestressed anchor which is anchored along the full length has no unbounded part to effectively transfer the pretension stress to the soil behind. Compared with the ordinary tension anchor, the effective range of compressive stress is smaller, and the effect of active support is poor;

(v)  The influence of pretension stress on the internal force of soil nails differs along the full length. The internal force close to the end declines rapidly, decreases gradually along the full length, and reduces to the minimum on the nail tail.

## 9. Recommendations and Future Research

According to the above conclusions, irregular construction methods, such as "tensioning-lagging of the anchor" and "soil predisturbance", have a great impact on the horizontal lateral displacement of the foundation pit and the expected effect of anchors. When soil nailing with prestressed anchors is designed or constructed, it is necessary (i) to ensure "supporting while digging, layered excavation and prohibiting over-excavation"; and (ii) to realize site safety management, monitor stress and displacement regularly, and establish an early warning safety mechanism. It is suggested to strengthen the management and monitoring during construction and require on-site workers to construct according to technical engineering specifications, which is of great significance to ensure project safety and achieve the purpose of economic efficiency.

**Author Contributions:** H.W.: writing—original draft preparation; J.C.: investigation; H.L.: data curation; Z.D.: writing—review and editing; B.C.: data curation. All authors have read and agreed to the published version of the manuscript.

**Funding:** This research was funded by (1) National Natural Science Foundation of China, grant number U1810203. (2) Ph.D. Foundation of Henan Polytechnic University, grant number B2018-67.

**Acknowledgments:** The writers wish to acknowledge the financial support of this research by the Comprehensive Design and Research Institute, Zhengzhou University. The kind assistance and valuable contributions of the staff of the School of Civil Engineering, Zhengzhou University and the generosity of Henan No. 7 Building Engineering Group who provided the site for the test, are gratefully acknowledged. Special thanks to Yuancheng Guo and Tonghe Zhou.

**Conflicts of Interest:** All authors declare that they have no conflict of interest.

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
