# Peer review of "Full-Scale Field Test on Construction Mechanical Behaviors of Retaining Structure Enhanced with Soil Nails and Prestressed Anchors"

_applsci, doi:10.3390/app11177928_

Round 1
Reviewer 1 Report
Line 10 Replace “found good” with a technical term.
Lines 10-12 Rewrite the sentence to clarify the importance and novelty of the research. These both are not apparent in the abstract.
Line 15 Add In particular or specifically
Line 17 Delete the word particularly
Line 18 add “and” before numbering iii)
Lines 27-30 should not be at the end of the abstract. Reconsider their position or rewrite.
Lines 36-37 Rearrange the position of e.g. “The advantages of soil-nailing include e.g. …..” Do this to all that applies
Line 46 -47 I don’t understand what e.g. are doing here. You need to give the examples, not just write the references that discusses them
Line 47 – 48 clarify the circumstances
Line 69 What were the precautions? Write them.
Lines 73 – 79 Why are you providing conclusions here? Reconsider these lines.
Line 81 Change to The tested foundation pit was designed or is designed
Lines 91-92 Change figure position and put it after the final paragraph of section 2
Line 108 Remove word separately
Table 1 Remove “O” from number 10
Lines 115-121 There is an unnecessary gap here between the table and the note
Line 143 Add dimensions in Figure 3 to demonstrate most of the characteristics of the structure described above the figure
Lines 152 – 166 There are unnecessary gaps. Remove line numbers from the left-hand side of Table 2 and Table 3
Line 174 Fix title of section 5
Line 176 Provide references for the equipment. This would help future researchers to use these instruments again if needed
Line 242 Why 3 months? Justify
From line 246 Remove numbering i), ii)….. and fix the quality of the figures
Very good analysis and discussion
Add recommendations and future research
The text in figures 3-7 is not visible. Change font size and formatting.
The quality of the figures needs fixing.
Design or construction procedure recommendations are necessary for this paper as the results have e direct effect on in-situ applications.
Reviewer 2 Report
The paper "Full-Scale Field Test on construction mechanical behaviors of Retaining Structure Enhanced with Soil Nails and Prestressed Anchors" reports the results of a series of in-situ tests conducted in order to analyze the mechanical behavior of retaining structure enhanced with soil nails and prestressed anchors. In particular, the attention is focused on the influence of different important design parameters as the effect of the prestressing force and the unbonded length of the prestressed anchors on the retaining system performance considering a real case study. The results obtained present different original aspects considering what has already reported in literature but the form of the manuscript needs an important revision before it can be published. In particular:
- it is necessary to improve the quality of the figures in all their aspects (image qualityl, legend, lettering, ecc.)
- verify and rewrite the Table
- in the note (line 12) it is opionion of the reviewer that φ represents the internal friction angle and τ the shear stress
- rewrite the references according to the journal guidelines and in the sentance "In addition, previous studies were mainly focused on the bahavior of such retaining structure installed in dense materials" consider as reported in M. Zucca, M. Valente. "On the limitations of decoupled approach for the seismic behavior evaluation of shallow multi-propped undergorund structures embedded in granular soils", Enginerring Structures 2020, 211, 110497
